# The Role of Cardiac Fibrosis in Diabetic Cardiomyopathy: From Pathophysiology to Clinical Diagnostic Tools

**DOI:** 10.3390/ijms24108604

**Published:** 2023-05-11

**Authors:** Kuo-Li Pan, Yung-Chien Hsu, Shih-Tai Chang, Chang-Min Chung, Chun-Liang Lin

**Affiliations:** 1Division of Cardiology, Department of Internal Medicine, Chang Gung Memorial Hospital, Chiayi Branch, Chiayi City 613, Taiwan; pankuoli64@gmail.com (K.-L.P.); cst1234567@yahoo.com.tw (S.-T.C.); cmchung02@hotmail.com (C.-M.C.); 2College of Medicine, Chang Gung University, Taoyuan City 333, Taiwan; 3Heart Failure Center, Chang Gung Memorial Hospital, Chiayi Branch, Chiayi City 613, Taiwan; 4Department of Nephrology, Kidney and Diabetic Complications Research Team (KDCRT), Chang Gung Memorial Hospital, Chiayi Branch, Chiayi City 613, Taiwan; libra@cgmh.org.tw; 5Kidney Research Center, Chang Gung Memorial Hospital, Taipei 105, Taiwan; 6Center for Shockwave Medicine and Tissue Engineering, Department of Medical Research, Chang Gung Memorial Hospital, Kaohsiung City 833, Taiwan

**Keywords:** diabetic cardiomyopathy, cardiac remodeling, cardiac fibrosis, heart failure, reactive oxygen species, hyperglycemia, inflammation, cardiac myocytes

## Abstract

Diabetes mellitus (DM) is a chronic metabolic disorder characterized by hyperglycemia due to inadequate insulin secretion, resistance, or both. The cardiovascular complications of DM are the leading cause of morbidity and mortality in diabetic patients. There are three major types of pathophysiologic cardiac remodeling including coronary artery atherosclerosis, cardiac autonomic neuropathy, and DM cardiomyopathy in patients with DM. DM cardiomyopathy is a distinct cardiomyopathy characterized by myocardial dysfunction in the absence of coronary artery disease, hypertension, and valvular heart disease. Cardiac fibrosis, defined as the excessive deposition of extracellular matrix (ECM) proteins, is a hallmark of DM cardiomyopathy. The pathophysiology of cardiac fibrosis in DM cardiomyopathy is complex and involves multiple cellular and molecular mechanisms. Cardiac fibrosis contributes to the development of heart failure with preserved ejection fraction (HFpEF), which increases mortality and the incidence of hospitalizations. As medical technology advances, the severity of cardiac fibrosis in DM cardiomyopathy can be evaluated by non-invasive imaging modalities such as echocardiography, heart computed tomography (CT), cardiac magnetic resonance imaging (MRI), and nuclear imaging. In this review article, we will discuss the pathophysiology of cardiac fibrosis in DM cardiomyopathy, non-invasive imaging modalities to evaluate the severity of cardiac fibrosis, and therapeutic strategies for DM cardiomyopathy.

## 1. Introduction

Diabetes mellitus (DM) is a major global health issue, and its prevalence has been increasing steadily over the past few decades. According to the International Diabetes Federation (IDF), in 2021, the global prevalence of diabetes among adults was estimated to be 537 million, or 1 in 10 adults worldwide. DM is also associated with an increased risk of cardiovascular disease, including coronary artery disease, stroke, and heart failure [1]. Diabetes-related cardiovascular complications are a major cause of morbidity and mortality in people with diabetes. For example, according to the American Heart Association, adults with diabetes are two to four times more likely to die from cardiovascular disease than those without diabetes [2]. The risk of cardiovascular complications is further increased in people with diabetes who have other risk factors, such as high blood pressure, high cholesterol, and smoking.

DM is a major risk factor for the development of heart failure [3,4]. DM affects the heart in three ways: (1) coronary artery disease due to accelerated atherosclerosis; (2) cardiac autonomic neuropathy; and (3) diabetic cardiomyopathy. One of the ways is through diabetic cardiomyopathy, which is left ventricular (LV) myocardial dysfunction that occurs in patients with DM in the absence of coronary artery disease, hypertension, or valvular heart disease [4]. DM cardiomyopathy is a complex and multifactorial disease that involves myocardial interstitial fibrosis and LV dysfunction. Many studies showed that LV diastolic dysfunction is one of the earliest functional alterations seen in the course of DM cardiomyopathy [5,6,7,8]. Early diagnosis and management of heart diastolic dysfunction by using clinical diagnostic tools are essential for preventing the development of heart failure. Clinical diagnostic tools such as echocardiography, cardiac CT, and heart MRI can be used to evaluate early heart diastolic dysfunction in patients with DM cardiomyopathy. The pathophysiology of DM cardiomyopathy involves myocardial interstitial fibrosis, which contributes to LV anatomic and functional remodeling. Interstitial extracellular matrix deposition promotes myocardial interstitial fibrosis and impairs LV compliance, leading to heart failure with preserved ejection fraction (HFpEF). Therefore, understanding the mechanisms of cardiac fibrosis in DM is crucial for developing effective therapeutic interventions. Several cellular effectors and bio-molecular mechanisms have been identified as mediators of myocardial fibrotic responses to DM. These include advanced glycation end-products (AGEs), oxidative stress, inflammation, and profibrotic growth factors. This review article discusses the clinical diagnostic tools for evaluating the organic and function remodeling mediated by DM cardiomyopathy, as well as the pathophysiologic myocardial fibrotic responses to DM, focusing on the organic and functional consequences of cardiac fibrosis, the cellular effectors, the bio-molecular mechanisms, and the possible therapeutic interventions.

## 2. Pathologic Myocardial Fibrosis of Diabetic Cardiomyopathy

There are generally considered to be two main types of myocardial fibrosis. One is replacement fibrosis, which is characterized by the replacement of dead or damaged myocardial tissue with collagen-rich scar tissue. Replacement fibrosis is typically associated with myocardial infarction and is often considered a reparative response to tissue injury. The other is interstitial fibrosis, which involves the expansion of the extracellular matrix (ECM) within the myocardium. Interstitial fibrosis is associated with a variety of pathological conditions, including hypertension, diabetes, and heart failure, and is a prevalent pathologic feature in major types of myocardial fibrosis. It is characterized by the deposition of ECM proteins in the cardiac interstitium, with activated fibroblasts and myofibroblasts being the central cellular effectors in cardiac fibrosis [9]. While apoptosis of cardiomyocytes is a contributing factor, progressive myocardial interstitial fibrosis also increases myocardial stiffness and reduces ventricular compliance, which may play a vital role in the development of LV diastolic dysfunction and the pathogenesis of DM cardiomyopathy with preserved LV ejection fraction [10,11]. The exact mechanisms underlying the development of pathologic myocardial fibrosis in diabetic cardiomyopathy are not fully understood, but it is thought to involve a complex interplay between various factors, including hyperglycemia, oxidative stress, inflammation, and alterations in cardiac metabolism. It is different from the mechanism of diabetic macrovasculopathy through the difference in vessel smooth muscle cells, and the basis for accentuated cardiomyocyte conversion to fibroblasts through the endothelial to mesenchymal transition (EndMT) in diabetic cardiomyopathy remains unclear. However, collagen deposition due to the activation of the ECM-synthetic program through cardiac fibroblasts without myofibroblast conversion has been reported in a diabetic mice animal model study [12]. Resident cardiac fibroblasts remain the predominant ECM-producing cells in diabetic cardiomyopathy. There are three major pathways contributing to this process, including the direct activation of cardiac fibroblasts through the transforming growth factor beta (TGF-β) and hyperglycemia [13], as well as the accumulation of advanced glycation end-products (AGEs) under hyperglycemia stress promoting the crosslinking of collagen fiber to stimulate cardiac fibroblasts [14,15,16]. This can lead to the development of fibrosis and stiffness in the heart muscle. Additionally, alterations in cardiac metabolism, such as glucose-independent pathways such as adipokines, endothelin-1 (ET-1) shifting towards the use of fatty acids for energy production, and neurohumoral pathways, can also activate cardiac fibroblasts, contributing to the development of pathologic myocardial fibrosis [15]. This can lead to the accumulation of toxic byproducts such as lipid peroxides, which can cause damage to cardiac cells and promote the production of fibrous tissue in the heart. Furthermore, oxidative stress, which results from an imbalance between the production of reactive oxygen species (ROS) and the body’s ability to neutralize them with antioxidants, can also contribute to the development of pathologic myocardial fibrosis. ROS can directly damage cardiac cells and activate signaling pathways that lead to the production of fibrous tissue in the heart [17]. Overall, a better understanding of the mechanisms underlying myocardial fibrosis and the involvement of cardiac fibroblasts in diabetic cardiomyopathy is critical to the development of effective therapeutic interventions. By targeting these pathways, it may be possible to prevent or slow the progression of myocardial fibrosis and improve cardiac function in patients with diabetic cardiomyopathy. (Figure 1).

### 2.1. Hyperglycemia Induces Cardiac Fibrosis

Cardiac fibrosis is a well-documented complication in patients with chronic DM [18]. High levels of glucose in the blood have been shown to be a significant contributor to DM-associated cardiac fibrosis, as demonstrated both in vitro and in animal studies. Specifically, when cardiac fibroblasts are cultured in a high-glucose environment, they tend to synthesize excessive amounts of ECM proteins such as collagens, fibronectin, and matricellular macromolecules [19,20,21]. In animal models, the administration of antihyperglycemic medications has been found to markedly attenuate myocardial fibrotic changes [22]. While poor sugar control is typically associated with evidence of organ fibrosis in human patients [23,24], some experimental and clinical studies have suggested that intensive glycemic control may not be sufficient to abolish cardiac fibrosis entirely [25]. Nonetheless, efforts to achieve optimal glucose control remain an essential component of DM management and may help to mitigate the risk of developing cardiac fibrosis.

### 2.2. Activating TGF-β Signaling Pathway

The TGF-β pathway is activated when TGF-β ligands bind to type II TGF-β receptors on the cell surface, which then recruit and activate type I receptors. The activated type I receptors then phosphorylate receptor-regulated Smads (R-Smads), such as Smad2 and Smad3. Phosphorylated R-Smads then form complexes with a common mediator Smad (co-Smad), Smad4, and translocate into the nucleus, where they regulate the transcription of target genes. In the nucleus, the Smads interact with transcription factors and co-regulators to regulate gene expression. TGF-β, a highly pleiotropic mediator, has been implicated in the pathogenesis of tissue fibrosis to a great extent [26]. Various cell types, such as fibroblasts, macrophages, epithelial cells, vascular cells, platelets, and organ-specific parenchymal cells, may be stimulated to produce and secrete TGF-β when exposed to high levels of glucose. TGF-β has been implicated in the development and progression of DM cardiomyopathy. Animal models of type 1 and type 2 diabetes have reported increased expression of TGF-β in cardiac myocytes, which is associated with cardiac fibrosis [27]. The TGF-β signal expressed in DM cardiac fibrosis may be due to the direct action of high glucose on TGF-β secretion and activation [28] or may be through angiotensin II pathway activation [29]. Prolonged exposure to aldose sugars can lead to non-enzymatic glycation and oxidation of proteins and lipids, resulting in the formation of AGEs. The accelerated accumulation of AGEs may mediate inflammation and fibrosis in diabetic tissues. AGEs may bind to the receptor for AGE (RAGEs), which are cell surface receptors that modulate the cellular phenotype. In fibroblasts, AGE/RAGE signaling stimulates the expression of matrix proteins and fibroblast proliferation [30]. The profibrotic effects of RAGE may be mediated, at least in part, through TGF-β [31]. In summary, TGF-β and AGEs are two critical factors that contribute to fibrosis in diabetic tissues. They can lead to the increased expression of matrix proteins, fibroblast proliferation, and modulation of the cellular phenotype, ultimately resulting in tissue fibrosis. Dysregulation of TGF-β signaling through the Smad3/Smad4 pathway plays a central role in promoting fibroblast activation and ECM synthesis, leading to the accumulation of ECM proteins in diabetic cardiomyopathy.

### 2.3. Activating Adipokines and ET-1 Signaling Pathway

Leptin, an adipokine secreted by adipose tissue, has been implicated in the development of cardiac fibrosis in a type 2 DM rat model by activating both cardiomyocytes and cardiac fibroblasts [32]. In contrast, adiponectin, another adipokine with anti-inflammatory, cardioprotective, and anti-atherogenic properties, has been shown to regulate cardiac fibrosis [33]. In fact, adiponectin has been found to exert anti-fibrotic effects in a model of angiotensin-induced cardiac remodeling in vivo [34]. However, the relative significance of the cellular actions between leptin and adiponectin on cardiomyocytes and fibroblasts in vivo remains unknown.

The ET-1 signaling pathway is another mechanism implicated in the pathogenesis of obesity-related cardiovascular diseases. ET-1 is a profibrotic peptide produced by vascular endothelial cells in response to cytokines, angiotensin II, or hypoxia. Experimental evidence has shown that ET-1 expression is enhanced in DM experimental models [35]. Endothelial cell-specific loss of ET-1 has been shown to attenuate myocardial fibrosis and reduce endothelial to mesenchymal transdifferentiation in streptozotocin-induced diabetic mice [35].

### 2.4. ROS-Dependent Microvascular Inflammation

ROS stands for reactive oxygen species, which are highly reactive molecules that can cause damage to cells and tissues. In DM cardiomyopathy, ROS can play a significant role in the development and progression of the disease. Previous studies have provided evidence for increased levels of ROS in animal models of DM [36]. In diabetic animals, hyperglycemia and insulin resistance are known to contribute to the generation of ROS in the heart, leading to oxidative stress. Oxidative stress occurs when there is an imbalance between ROS production and the body’s antioxidant defense mechanisms. ROS can damage cellular structures, including proteins, lipids, and DNA, leading to cellular dysfunction and death. ROS can also impair the function of mitochondria, which are the energy-producing structures within cells, leading to further cellular dysfunction. Furthermore, ROS are believed to play a significant role in cardiac fibrogenesis mediated by angiotensin II, cytokines, and growth factors. There are several strategies that can be used to reduce ROS levels including lifestyle changes, such as exercise and a healthy diet, as well as the use of antioxidant supplements or medications. Pharmacological interventions targeting oxidative stress have been shown to effectively reduce cardiac interstitial fibrosis in DM models [37]. There are numerous studies that have investigated the effects of pharmacological inhibitors targeting oxidative stress on myocardial interstitial fibrosis in diabetic cardiomyopathy. There were some studies that have shown significant reductions in myocardial interstitial fibrosis. Epigallocatechin gallate (EGCG), a natural antioxidant, reduced high glucose-induced oxidative stress and inhibited cardiac interstitial fibrosis in a diabetic rat model [38]; Curcumin, a natural polyphenol, reduced oxidative stress and inhibited Akt/GSK-3β-induced fibrosis in a diabetic rat model [39]; and N-acetylcysteine (NAC), a glutathione precursor, reduced high glucose-induced oxidative stress and inhibited interstitial fibrosis in a diabetic mice model [40]. Melatonin, a natural antioxidant, and Resveratrol, a natural polyphenol, also reduced oxidative stress and inhibited myocardial interstitial fibrosis in a diabetic rat model [41]. These are just a few examples of the many studies investigating pharmacological inhibitors targeting oxidative stress and myocardial interstitial fibrosis in diabetic cardiomyopathy. It should be noted that the efficacy and safety of these inhibitors should be carefully evaluated before clinical use.

### 2.5. ECM Receptors Mediating DM Cardiomyopathy

ECM receptors including integrins, syndecans, discoidin domain receptors (DDRs), hyaluronic acid receptors, laminin receptors, and matrix metalloproteinases (MMPs) are expressed on the surface of cells and mediate their interaction with the ECM [42]. ECM receptors and their associated signaling pathways have been shown to play a role in the development and progression of DM cardiomyopathy [42]. One of the key ECM receptors involved in DM cardiomyopathy is the RAGE [30]. In addition to RAGE, other ECM receptors, such as integrins and DDR2, have also been implicated in DM cardiomyopathy. Integrin-mediated signaling has been shown to regulate cardiac fibroblast activation and ECM synthesis in response to high glucose levels [43]. DDR-mediated signaling has been shown to promote cardiac fibroblast differentiation into myofibroblasts and ECM synthesis in response to increased oxidative stress [43]. The signaling pathways activated by ECM receptors in diabetic cardiomyopathy can lead to various pathological changes in the heart muscle, such as increased fibrosis, inflammation, and oxidative stress. Cardiac fibrosis, in particular, is a hallmark of diabetic cardiomyopathy and is characterized by the accumulation of ECM proteins, such as collagen, in the heart tissue. Cardiac fibrosis can lead to stiffening of the heart muscle, impairing its ability to pump blood effectively.

## 3. Clinical Tools for Cardiac Functional and Anatomic Remodeling Diagnosis in DM Cardiomyopathy

DM leads to myocardial cell injury, interstitial fibrosis, and, ultimately, ventricular systolic and diastolic dysfunction, which mediate diabetic cardiomyopathy to become HFpEF clinically [44]. Cardiac functional and anatomic remodeling are key features of diabetic cardiomyopathy, and their timely diagnosis is crucial for the effective management of the disease. Several clinical tools are available to diagnose cardiac remodeling in diabetic patients. One commonly used tool is echocardiography, which is non-invasive and provides detailed information about cardiac function and structure. Heart CT, cardiac MRI, and nuclear imaging are other valuable tools for diagnosing cardiac remodeling in DM. They provide high-resolution images of the heart, allowing for accurate assessment of cardiac structure and function. They can detect changes in myocardial mass, wall thickness, and perfusion, which are indicative of cardiac remodeling in DM. Dilated LA, marked decreased early (E)/late (A) mitral flow velocity ratio, and marked increased E/mitral annular e’ velocity ratio are the earlier parameters to present DM diastolic cardiomyopathy. Moreover, they can be detected by those non-invasive examination tools (Figure 2).

### 3.1. Echocardiography

Traditionally, Doppler echocardiography is used to assess mitral inflow velocity curves and measure LV size in order to diagnose and classify the degree of LV diastolic dysfunction [45,46]. The American Society of Echocardiography recently updated their recommendations for evaluating LV diastolic function, which now includes parameters such as mitral annular e’ velocity, E/e’ ratio, LA maximum volume index, and the initial recommended approach of using the mitral inflow E/A ratio [47]. If a patient has clinical symptoms of heart failure with an E/A ratio ≥ 2 measured by Doppler echocardiography, the diagnosis of LV diastolic dysfunction should be established. However, in young individuals, an E/A ratio > 2 can be a normal variant, so it is important to look for other symptoms and signs of heart failure [48]. For patients with a mitral inflow E/A ratio between 0.8 and 1.9, or with a ratio ≤ 0.8 combined with peak E velocity > 50 cm/s, LA volume index (LAVI) > 34 mL/m^2^, peak velocity of tricuspid regurgitation > 2.8 m/s measured by CW Doppler echocardiography, and mitral average E/e’ ratio > 14, further confirmation is needed to determine the elevated LV filling pressure and diagnose LV diastolic dysfunction in this patient group [49,50]. Recent developments in 2D speckle tracking echocardiography have allowed for the addition of new parameters to assess LV diastolic dysfunction, such as an LA contractile strain rate < −1.66/s, which indicates decreased LA pump function and improves the accuracy of LV diastolic dysfunction diagnosis in patients with diabetic cardiomyopathy [51]. In conclusion, the evaluation of LV diastolic function involves multiple parameters and requires careful interpretation of echocardiography results. It is important for clinicians to keep up to date with the latest recommendations in order to accurately diagnose and manage LV diastolic dysfunction in their patients.

### 3.2. Cardiac CT

Compared to 2D echocardiography, which can evaluate LV diastolic functional parameters including the E/A ratio, E/e’, etc., to diagnose and class the degree of LV diastolic dysfunction, the cardiac CT mainly evaluates the anatomic remodeling such as dilated LA or dilated LV in patients with LV diastolic dysfunction. A previous study showed that high-degree LV diastolic dysfunction would mediate higher LA pressure leading to LA enlargement [52]. The study of Kaiume, M et al. showed that a maximum anteroposterior diameter/maximum medial diameter of the thorax > 0.165, a maximum anteroposterior diameter of the LA > 43.9 mm, and a maximum transverse diameter of the heart can be the indicators for LV diastolic dysfunction [53]. Clinically, DM cardiomyopathy should be considered in DM patients with enlarged LA. Moreover, as we know, the LA total emptying fraction (LATEF) is a measure of global LA function. The study by Lessick, J. et al. showed that LATEF < 40%, detected by cardiac CT, can accurately detect advanced LV diastolic dysfunction and has additive value to echocardiography-derived diastolic dysfunction [54]. A cardiac CT scan is a valuable tool in screening for LV diastolic dysfunction, providing a more comprehensive evaluation of organic remodeling and offering important diagnostic indicators such as LATEF and LA measurements.

### 3.3. Cardiac MRI

Cardiac MRI has powerful diagnostic utility in heart muscle diseases such as Fabry disease and amyloidosis. Currently, cardiac MRI is also the gold-standard technique to evaluate LV mass and volumes and provides superior reproducibility compared with echocardiography [55,56]. As we know, Doppler echocardiography and tissue Doppler imaging can measure the mitral inflow velocities and myocardial velocities to classify the degree of LV diastolic function. Furthermore, a newly developed cardiac MRI golden-angle method allows the acquisition of 150 to 250 frames per cardiac cycle to match that of echocardiography, which also can measure the mitral inflow and pulmonary inflow velocities by phase-contrast MRI to diagnose the LV diastolic dysfunction [57]. Phase-contrast cardiac MRI also can be used to determine myocardial velocities, and cardiac MRI-derived mean e’ and E/e’ have consistently shown excellent correlation with the values obtained from Doppler echocardiography and tissue Doppler imaging in patients with LV diastolic dysfunction [58,59]. Progressive enlarged LA is a hallmark of elevated LV filling pressures in patients with severe LV diastolic dysfunction [60]. Cardiac MRI provides a more accurate measurement of LA size compared with echocardiography and could evaluate LA ejection fraction (LAEF) correctly. Increased LA volumes, reduced LAEF, reduced LA reservoir, and booster pump strains are all associated with LV diastolic dysfunction [61]. Different from cardiac CT, cardiac MRI has the capability to detect both focal myocardial fibrosis using late gadolinium enhancement (LGE) and diffuse interstitial fibrosis through parametric mapping sequences by using native T1 and extracellular volume quantification.

### 3.4. Nuclear Imaging

There is a two-fold risk of silent myocardial infarction and mortality in DM patients with autonomic neuropathy compared to DM patients without autonomic neuropathy [62]. Traditionally, the evaluation and management of the patients affected by HF with LV systolic or diastolic dysfunction have been focused on hemodynamic abnormalities and organic abnormality, but neurohormonal and molecular pathophysiologies have gained attention along with the rapid development of nuclear medicine instruments and the widespread availability of new radiopharmaceutical agents. Cardiac sympathetic neuroimaging studies have found decreased 123I-MIBG-derived radioactivity in diabetic patients compared to controls [63]. It was also shown that cardiac automatic neuropathy was an independent risk marker for the presence of LV diastolic dysfunction in patients with DM. The activation of the sympathetic nervous system contributes to LV diastolic dysfunction. Early diagnosis and treatment of cardiac automatic neuropathy are advocated for preventing LV diastolic dysfunction in patients with DM cardiomyopathy [62].

## 4. Current Treatment to Reverse Diabetic Cardiac Remodeling

Diabetic cardiac remodeling refers to the structural and functional changes that occur in the heart as a result of diabetes. It can lead to a variety of complications, including heart failure, arrhythmias, and myocardial infarction. Currently, the best approach to reversing diabetic cardiac remodeling is to manage the underlying diabetes through lifestyle changes and medication. This can include regular exercise, a healthy diet, and medications such as insulin or oral hypoglycemic agents. In addition, controlling blood pressure and cholesterol levels is also important to reduce the risk of cardiovascular disease. However, it is important to note that reversing diabetic cardiac remodeling can be challenging and often requires a multi-faceted approach.

Cardiac fibrosis, major cardiac remodeling in DM cardiomyopathy, reduces LV compliance, contributing to clinical HFpEF, which has increased mortality and a higher incidence of hospitalizations [64]. Early detection and treatment of diabetic cardiomyopathy are important for preventing the development and progression of pathologic myocardial fibrosis. It was also shown that ventricular fibrosis increased the risk of ventricular arrhythmias and sudden death observed in DM patients [65]. Otherwise, DM cardiomyopathy may also induce atrial fibrosis, increasing the incidence of embolic stroke. As we know, poor glycemic control is associated with a high incidence of heart failure [66]. Traditionally, tight glycemic control may be effective in the attenuation of cardiac fibrosis. Unfortunately, the number of cardiovascular events, risk of heart failure, and incidence of atrial fibrillation were not significantly lower among DM patients using intensive glycemic control methods clinically [67,68,69]. Even relations between outcome and effects on cardiac fibrosis have not been studied. Clinical evidence suggests that DM has an increased incidence of post-infarction heart failure predominantly due to diastolic dysfunction [70]. It is pointed out that cardiac-fibrosis-mediated LV diastolic dysfunction may play an important role in DM cardiomyopathy and could be a therapeutic target. Our review showed that non-invasive examination tools including echocardiography, heart CT, cardiac MRI, and nuclear imaging can evaluate the severity of anatomic remodeling and functional remodeling in patients with DM cardiomyopathy. Those tools can not only be used for the diagnosis of DM cardiomyopathy but also for follow-up to evaluate the efficacy of clinical therapy. Clearly, additional pharmacologic strategies are needed to inhibit and reverse cardiac fibrosis and prevent the development of DM cardiomyopathy. Established pharmacologic therapy for HFrEF to reverse LV remodeling through inhibiting RAAS by using ACE inhibitors, angiotensin receptor blockers, and aldosterone antagonists may be also valuable to safely treat HFpEF mediated by DM cardiomyopathy. Whether these approaches exert beneficial actions through the attenuation of cardiac fibrosis in DM cardiomyopathy is unclear and needs further investigation. A meta-analysis of several studies found that TGF-β levels are significantly increased in the serum and myocardium of individuals with DM cardiomyopathy compared to individuals without diabetes. The study suggests that TGF-β may be a useful biomarker for the diagnosis and prognosis of DM cardiomyopathy. The study of Pang et al. found that TGF-beta inhibition could attenuate myocardial fibrosis and improve cardiac function in a rat model of DM cardiomyopathy. The study suggests that targeting TGF-beta signaling could be a promising therapeutic approach for this condition [71]. The study by Liu et al. found that treatment with an antioxidant compound, N-acetylcysteine, improved cardiac systolic and diastolic function and reduced oxidative stress markers. Strikingly, N-acetylcysteine treatment, which had earlier and longer treatment, produced significant improvement in cardiac function and less fibrosis in a diabetic mice animal model [40]. The formation of AGEs on extracellular matrix components leads to accelerated increases in collagen crosslinking that contribute to myocardial stiffness in diabetes. The study by Candido showed that cleavage of preformed AGE crosslinks with ALT-711 leads to the attenuation of diabetes-associated cardiac abnormalities in rats [72].

Clinically, further novel pharmacologic strategies to eliminate cardiac fibrosis by blocking the activation of the ROS system, AGE-mediated ET-1, or the TGF-β pathway induced by hyperglycemia in DM cardiomyopathy are still under investigation.

## 5. Conclusions

DM cardiomyopathy is a condition that occurs in individuals with diabetes and is characterized by abnormalities in the structure and function of the heart. The exact mechanisms underlying the development of DM cardiomyopathy are not fully understood, but it is thought to involve a complex interplay between various factors, including hyperglycemia, insulin resistance, inflammation, oxidative stress, and alterations in cardiac metabolism.

Diastolic dysfunction is the key feature of DM cardiomyopathy. Over time, it can progress to systolic dysfunction and can lead to heart failure. Non-invasive imaging techniques, such as echocardiography, cardiac CT, cardiac MRI, and nuclear imaging, can be used to accurately evaluate the severity of diastolic dysfunction and the degree of cardiac fibrosis in DM cardiomyopathy.

Myocardial interstitial fibrosis is a significant contributor to LV anatomic and functional remodeling in DM cardiomyopathy. The activation of the TGF-β pathway, inflammation, and oxidative stress play a key role in the development of myocardial interstitial fibrosis. Chronic inflammation, which is often present in individuals with diabetes, can lead to the release of inflammatory cytokines and the activation of immune cells in the heart. This can cause damage to cardiac cells and impair their function. Similarly, oxidative stress, which results from an imbalance between the production of ROS and the body’s ability to neutralize them with antioxidants, can lead to damage to cardiac cells and impaired cardiac function. ROS can directly damage cellular structures such as proteins, lipids, and DNA and can also activate signaling pathways that lead to inflammation and cell death.

In addition to tight glycemic control, reversing cardiac fibrosis has emerged as a promising precision therapeutic strategy for DM cardiomyopathy. Various pharmacological agents, such as angiotensin-converting enzyme (ACE) inhibitors, ARBs, and aldosterone antagonists, have been shown to have anti-fibrotic effects and improve LV diastolic function in patients with DM cardiomyopathy. In summary, evaluating the degree of cardiac fibrosis using non-invasive imaging techniques and reversing cardiac fibrosis using precision therapeutic strategies that eliminate cardiac fibrosis by blocking the activation of the ROS system, AGE-mediated ET-1, the ECM receptor pathway, or the TGF-β pathway induced by hyperglycemia in DM cardiomyopathy can help improve LV diastolic function and prevent the progression of HFrEF in patients with DM cardiomyopathy (Figure 3).

## Figures and Tables

**Figure 1 ijms-24-08604-f001:**
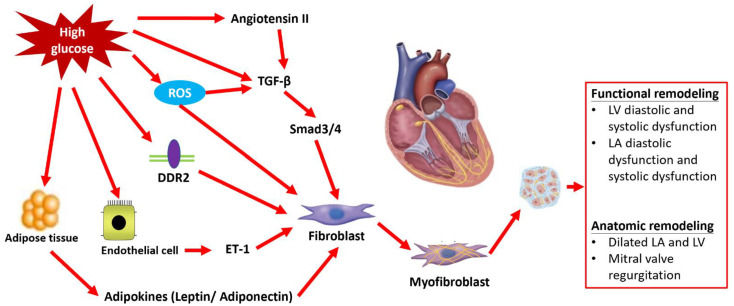
Hyperglycemia activates four major pathways including (1) transforming growth factor beta (TGF-β) cascades directly and through angiotensin II pathway; (2) reactive oxygen species (ROS)-dependent microvascular inflammation process; (3) Discoidin Domain Receptor 2 (DDR2) signaling; (4) Endothelin-1 (ET-1) signaling; and (5) Adipokines pathway to promote the activation of cardiac fibroblasts to increase progressive myocardial interstitial fibrosis mediating left ventricular (LV) and left atrial (LA) functional remodeling in the pathogenesis of diabetes mellitus (DM) cardiomyopathy with preserved LV ejection fraction.

**Figure 2 ijms-24-08604-f002:**
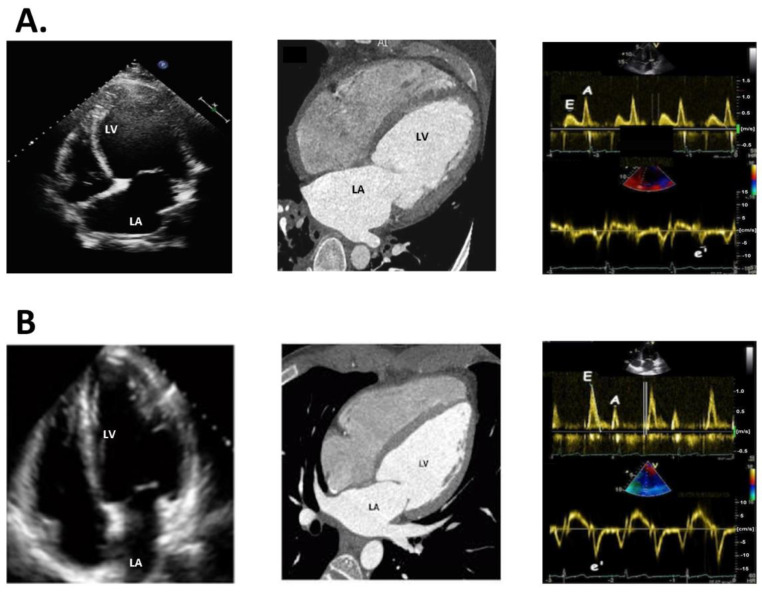
Diabetes mellitus (DM) cardiomyopathy mediates left ventricular (LV) diastolic dysfunction and dilated left atrium (LA), becoming heart failure with preserved ejection fraction (HFpEF) clinically. Panel (**A**) represents DM cardiomyopathy with dilated LA and LV diastolic dysfunction including marked decreased early (E)/late (A) mitral flow velocity ratio and marked increased E/mitral annular e’ velocity ratio, measured by echocardiography and cardiac computed tomography (CT). Panel (**B**) represents a normal heart with normal size of LA and normal LV diastolic function with E > A, measured by echocardiography and CT.

**Figure 3 ijms-24-08604-f003:**
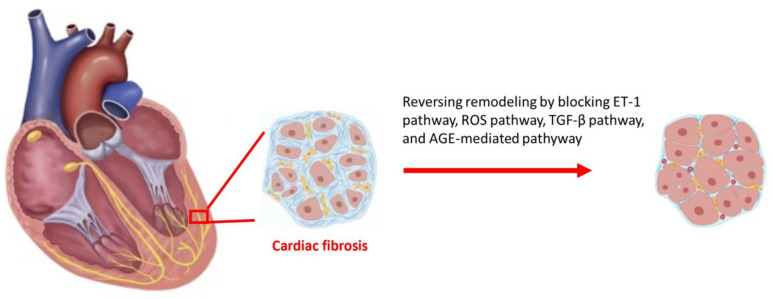
Reversing cardiac fibrosis can be a precision therapeutic strategy by blocking the activation of the reactive oxygen species (ROS) [38,39,40] system, advanced glycation end-products (AGE)-mediated [72], endothelin-1 (ET-1) [35], or the transforming growth factor beta (TGF-β) [71] pathway induced by hyperglycemia in diabetes mellitus (DM) cardiomyopathy.

## Data Availability

Not applicable.

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
