# Peer review of "The Role of Cardiac Fibrosis in Diabetic Cardiomyopathy: From Pathophysiology to Clinical Diagnostic Tools"

_ijms, 2023, doi:10.3390/ijms24108604_

Round 1
Reviewer 1 Report
The review article titled “The Role of Cardiac Fibrosis in Diabetic Cardiomyopathy: From Pathophysiology to Clinical diagnostic tools” by Kuo-Li et al., is a relevant topic in the field of Diabetic cardiomyopathy. The authors have done a good job in getting most of the pathophysiological aspects of the development of cardiac fibrosis under diabetes condition. However, I feel the review lacks inputs from very recent studies in the field of cardiac fibroblasts.
Major comments:
1) The role of extracellular matrix (ECM) receptors and their signaling in cardiac fibroblasts is completely devoid in the review paper. Metabolic syndrome / Hyperglycemia induces increased expression of collagen receptor, Discoidin Domain Receptor 2 (DDR2) and signaling through these receptors have been shown to activate cardiac fibroblasts and lead to profibrotic phenotype.
2) Redo Figure 1 to clearly indicate the activation and downregulation of the signaling pathways shown to easily comprehend the effects of high glucose on various effectors. Additionally, the meaning of the functional and organic remodeling depicted should be brought out in a better way by depicting the characteristic features of remodeling involved.
3) Figure 1 shows effector molecule Smad3 and does not discuss it in text section 2.2. “Activating TGF beta signaling pathway”. Smad4 is instead discussed in the text. Please include references to Smad3 also in the text. TGF beta signaling involves both the Smad3 and Smad4.
4) Reference to leptin studies is not given in the text section 2.3.
5) Try to redo Figure 2 to compare normal heart images and Diabetic cardiomyopathy hearts using various clinical imaging tools to better appreciate the impact of the tools
6) Figure 3 needs improvement along with additional references of studies that have done blocking of ROS, AGE, ET-1 and TGF-beta pathways. This also needs to be added to the text.
Overall, the review needs more specific studies added to improve the quality of the review. Additionally, include more recent studies in the field to make it more up to date.
Minor comments:
1) There are typo-errors in line 65-67.
2) Rewrite line 76. “In contrast to dedifference of vessel smooth muscle cells…” the sentence meaning is not clear.
3) Lines 89-97, has some truncations and loss of clarity of the sentence. “Additionally, alterations in cardiac metabolism….” Please re-write it.
4) Line 322, “Unfortunately, There…” correct to “Unfortunately, there…” remove capital “T”
5) Line 330-335, please rewrite. The sentence clarity is missing.
The paper needs English language edits to improve the ease of reading and clarity of thoughts.
Reviewer 2 Report
This paper proposed by Pan et al, entitled “The Role of Cardiac Fibrosis in Diabetic Cardiomyopathy From Pathophysiology to Clinical diagnostic tools” focused on DM cardiomyopathy. The pathophysiologic myocardial fibrotic responses to DM are discussed, with an emphasis on the organic and functional consequences of cardiac fibrosis, the cellular effectors, the biomolecular mechanisms, and potential therapeutic interventions. They also discuss the clinical diagnostic tools how to evaluate the organic and function remodeling mediated by DM cardiomyopathy. The review is interesting but there are several areas which need to be improved.
Major comments -
1. The abstract discusses how cardiovascular complications of DM are the major cause of morbidity and mortality in diabetic people. What is the global prevalence of the disease? Discussing the numbers will highlight the significance of the issue. The introduction needs to be strengthened.
2. The writers discuss MF is prevalent in many different types in line 65. Other kinds of Myocardial Fibrosis should be discussed.
3. Represent/ show all the molecules involved in the pathophysiology mechanisms of diabetic cardiomyopathy.
4. On lines 191-195, the authors discuss many strategies for reducing ROS levels, including the "use of antioxidant supplements or medications." What are the drugs used to reduce ROS levels, and if it is an FDA-approved drug, can authors describe it? (Cite the reference).
5. List all invitro and invivo pharmacological inhibitors (studies) that target oxidative stress and significantly reduce myocardial interstitial fibrosis in DM.
6. A more concise presentation of the conclusion section is required.
Minor comments -
1) Please double-check the spellings of interstitium and ctivated on line 67.
2) Remove the word "can" from line 91.
3) The whole manuscript needs language editing in order to make it easier to read and follow. Please, stay consistent with the style for references.
Extensive editing of the language is needed because of the numerous spelling errors.
Round 2
Reviewer 1 Report
The authors have improved the review article by including changes according to my comments. There are still some minor errors that I noticed in the new sections.
Minor comments:
1) In Figure 1, spell-check for "Anatolic".
2) In Figure 2, the legends do not say anything about group B (panel B). Be little more elaborate by saying what each panel represents.
Try to edit the English language better clarity. I am not sure about the usage of the term "Organic remodeling" in the article. "Tissue remodeling" is what I think would be correct usage or "remodeling at organ level".
Reviewer 2 Report
The authors have addressed all the comments. I have no comments further.
The authors have corrected the grammatical errors. The quality of the English language is good.
Author Response
Dear Reviewer
Thank you for your comments